# Human Breast Milk Contamination with Aflatoxins, Impact on Children’s Health, and Possible Control Means: A Review

**DOI:** 10.3390/ijerph192416792

**Published:** 2022-12-14

**Authors:** Noreddine Benkerroum, Amir Ismail

**Affiliations:** 1Expertise Aliments Santé, Food Health Consultancy, 7450 Dollier Str., Montréal, QC H1S 2J6, Canada; 2Institute of Food Science and Nutrition, Bahauddin Zakariya University, Multan 60000, Pakistan

**Keywords:** human breast milk, aflatoxin M1, carry-over, stunting, child growth and development, health risk, influx/efflux transport, contamination status, exposure

## Abstract

Aflatoxins are natural toxicants produced mainly by species of the *Aspergillus* genus, which contaminate virtually all feeds and foods. Apart from their deleterious health effects on humans and animals, they can be secreted unmodified or carried over into the milk of lactating females, thereby posing health risks to suckling babies. Aflatoxin M1 (AFM1) is the major and most toxic aflatoxin type after aflatoxin B1 (AFB1). It contaminates human breast milk upon direct ingestion from dairy products or by carry-over from the parent molecule (AFB1), which is hydroxylated in the liver and possibly in the mammary glands by cytochrome oxidase enzymes and then excreted into breast milk as AFM1 during lactation via the mammary alveolar epithelial cells. This puts suckling infants and children fed on this milk at a high risk, especially that their detoxifying activities are still weak at this age essentially due to immature liver as the main organ responsible for the detoxification of xenobiotics. The occurrence of AFM1 at toxic levels in human breast milk and associated health conditions in nursing children is well documented, with developing countries being the most affected. Different studies have demonstrated that contamination of human breast milk with AFM1 represents a real public health issue, which should be promptly and properly addressed to reduce its incidence. To this end, different actions have been suggested, including a wider and proper implementation of regulatory measures, not only for breast milk but also for foods and feeds as the upstream sources for breast milk contamination with AFM1. The promotion of awareness of lactating mothers through the organization of training sessions and mass media disclosures before and after parturition is of a paramount importance for the success of any action. This is especially relevant that there are no possible control measures to ensure compliance of lactating mothers to specific regulatory measures, which can yet be appropriate for the expansion of breast milk banks in industrialized countries and emergence of breast milk sellers. This review attempted to revisit the public health issues raised by mother milk contamination with AFM1, which remains undermined despite the numerous relevant publications highlighting the needs to tackle its incidence as a protective measure for the children physical and mental health.

## 1. Introduction

Breast milk is an invaluable food for the neonates and infants whom it provides with the essential macro- and micro-nutrients for normal growth and development [1]. It is also a unique source for the nursing children of passive immunity components (e.g., immunoglobulins, lymphocytes, cytokines, and growth factors) and antimicrobial agents (lysozyme, lactoferrins, and antimicrobial peptides) that stimulate their defense mechanisms against infections and inflammatory immunological diseases during the first days of life and even later [2,3]. For these main reasons, breastfeeding was recommended by the most influential health organizations, e.g., the World Health Organization (WHO), Center for Disease Control and Prevention (CDC), and the American Academy of Pediatrics (AAP), to be the exclusive diet for the first six months of life and then be continued with appropriate complementary foods for at least one year of age [4,5,6].

Nonetheless, despite the unrivaled nutritional and health benefits of breast milk to the nursing infants, it can also be a source of safety concerns, as it may contain various chemical toxicants (e.g., natural toxins, drug residues, pesticide residues, and heavy metals) deriving from the mother’s diet or medication [7,8,9]. Therefore, to ensure the wholesomeness of breast milk as the main source of nutrients and passive immunity transfer to nursing children, the mothers’ diet and medication should be stringently monitored to prevent or reduce to the lowest possible levels the excretion of chemical hazards in the milk. Among the natural toxicants that can be transferred from breast milk or colostrum to nursing children, mycotoxins, especially aflatoxins, are raising increased concerns. In fact, it is well established that human milk can be contaminated with various mycotoxins, including the conventional (e.g., aflatoxins, deoxynivalenol, and ochratoxin A) and the emerging (e.g., alternariol monomethyl ether and beauvericin) ones, which can occur individually or as mixtures of two or more [10,11,12,13]. However, despite the many publications highlighting the potential health risks posed to infants fed with aflatoxin-contaminated breast milk, this issue remains somewhat overlooked by the public health authorities, consumers, and nursing mothers themselves.

This review aims to highlight the recent advances on the occurrence of aflatoxins in breast milk and the hypothetical and real health risks they represent for nursing children. A special emphasis is put on aflatoxin M1 (AFM1) as the most prevalent and potent aflatoxin, after its parent aflatoxin B1 (AFB1), in milk of mammals, including human breast milk, which exposes not only mothers to significant health risks, but also the nursing children who belong to the high-risk group of consumers. Measures to prevent or reduce breast milk contamination with aflatoxins, including regulatory measures that can apply are also discussed.

## 2. Routes of Breast Milk Contamination with AFM1

### 2.1. Diet

The diet of the nursing mother appears to be the exclusive source for the contamination of breastfed children with aflatoxins; passive skin penetration and inhalation may have a minor contribution but there is no evidence to our knowledge for these routes of contamination. Upon ingestion with contaminated foods, aflatoxins are absorbed in the small intestine and directed through the portal bloodstream to the liver where they undergo various metabolic reactions mainly catalyzed by microsomal cytochrome P450 (CYP 450) oxidases [14]. The resulting intermediate metabolites then follow different pathways whereby they can either generate reactive intermediates, i.e., epoxides, that interact with DNA, RNA, and proteins to form toxic adducts or combine with soluble nucleophilic molecules, e.g., glutathione and glucuronic acid, to form non-toxic conjugates that are eliminated in biological excretions [15,16,17]. Alternatively, they are distributed unmodified via the systemic circulation to various tissues and biological fluids, including milk. Notably, the detoxification of AFB1 in the liver produces 4 major derivatives resulting from hydroxylation, demethylation, or reduction reactions (Figure 1). The hydroxylation of AFB1 furan ring by CYP1A2 isozyme generates AFM1 while the hydroxylation of its cyclopentenone ring by CYP3A4, CYP3A7, or CYP2A13 forms AFQ1; demethylation of AFB1 with CYP2A13 produces AFP1 and its ketoreduction by cytosolic nicotinamide adenine dinucleotide phosphate (NADPH)-dependent reductase yields aflatoxicol (AFL) [14].

Although the latter AFB1 derivatives are considered as detoxified metabolites of AFB1, they retain toxicities to various degrees since they conserve the double bond 8,9 of the furan ring being the main precursor of toxic adducts with DNA, RNA, and proteins upon epoxidation [14,18]. It is worth mentioning, however, that AFL, the most toxic AFB1 derivative (20–50% of AFB1 potency) [19], is a short-lived metabolite, which is readily converted back to the parent molecule and which appears to not react directly with DNA to form mutagenic adducts [20]. Indeed, despite its demonstrated ability to form DNA adducts in vitro in isolated rainbow trout hepatocytes, when dosed in vivo in fry rainbow trout, the DNA adducts formed were largely dominated by AFB1-DNA, suggesting that AFL is converted into AFB1 prior to its activation by epoxidation and the formation of DNA adducts [20,21]. Nonetheless, detection of AFL in milk from different mammals, including humans, at significant levels suggests that it is not completely converted into the parent aflatoxin during lactation and that it can exert toxicological effects on consumers in its own right [22]. On the other hand, AFQ1 and AFP1 are weekly toxic to non-toxic metabolites [14,23,24], and unlike AFM1 they are essentially excreted in urine and feces [25,26]; therefore they are of less concern to the mother and child health. Consequently, AFM1 remains the main toxic derivative of AFB1 that can contaminate human breast milk owing to its relatively high toxicity (retains up to 10% of AFB1 toxicity) [27,28,29], relative abundance among the AFB1 metabolites [30], and established carcinogenicity in animals [31]. In addition, it has a high affinity to the mammary glands making breast milk the preferred route of excretion, as it represents 95% of AFB1 metabolites excreted in milk [26,32]. Hence, the prime importance given to the contamination of milk, including human breast milk, with this aflatoxin from the scientific and public health standpoints. Therefore, this review will focus on AFM1 in breast milk and the health risks it poses to infants and young children. It should be emphasized, however, that the occurrence of other aflatoxins and mycotoxins in general in breast milk should not be disregarded as they can have their own toxicities and/or act in synergy with AFM1 or with each other to increase the health risks on the mothers and the nursing children. For example, AFL was shown to contaminate 13% of 290 analyzed bovine milk samples in Mexico at levels varying between <0.05 and 12.4 mg/L [33]. Another study on Oaxaca cheese marketed in Mexico city showed that 97% of 30 analyzed samples were contaminated with AFL at average level of 13.1 ng/g (range <0.01–25.5 ng/g) [22]. This is especially alarming that AFL has the same tumorigenic potency as its parent AFB1 [28] and warrants to be studied separately as a public health hazard in milks, including human milk. Occurrence of mycotoxin mixtures in breast milk with different toxicity potencies and to different levels and frequencies are well documented [11,12].

Apart from the contamination of breast milk with the AFM1 via the liver as an AFB1 carry-over metabolite, it can also be directly ingested with dairy products and distributed unmodified through the liver to breast milk via blood circulation (Figure 1). Moreover, AFB1 can also transit from the liver unmodified to the breast where it can be hydroxylated to AFM1 by the epithelial cells of the mammary glands (Figure 1), as was evidenced in vitro in the bovine epithelial cell line BME-UV1 where about 1% of the AFB1 was transformed into AFM1 most likely by a cytochrome enzyme of the CYP1A subfamily [18,30]. This appears to apply to the human mammary glands whose epithelial cells were reported to produce CYP1A1 isoform [8,34]. However, the actual bioconversion of AFB1 into AFM1 in the mammary glands and the enzyme(s) involved remain to be clarified, especially that CYP1A2 responsible for this bioconversion in the liver and in other tissues was not detected in the mammary epithelial cells [35]. Moreover, the capacity of bovine mammary epithelial cells to convert AFB1 into AFM1 was shown to be six-fold weaker than that of the hepatocytes [36], suggesting that the mammary glands would have a minor contribution to the accumulation of AFM1 in milk.

It appears from the above discussion, that AFM1 may contaminate breast milk at toxic levels for the mother and her child, especially when the dietary exposure of the nursing mother to AFB1 or AFM1 is too high. Although the extent of breast milk contamination with AFM1 is dictated by its direct intake from dairy products as well as by AFB1 carry-over, the latter means would represent the major route in countries where the diet is characterized by a high consumption pattern of foods prone to AFB1 contamination (e.g., cereals, dry fruits, spices, and nuts) [37]. Additionally, the contribution of AFB1 carry-over to the extent of breast milk contamination with AFM1 is dependent on the carry-over rate (COR), i.e., the proportion of AFB1 converted into AFM1 in the liver and possibly in the mammary glands. While the carry-over rate has been extensively studied in lactating domestic animals exposed to different amounts of aflatoxins [22,38,39,40,41], such studies are lacking in lactating mothers due the ethical considerations severely restricting the use of human subjects in experiments where they can be intentionally given aflatoxin-contaminated foods [42,43,44]. Nevertheless, the correlation between AFB1 dietary intake and human milk contamination with AFM1 is well established [32,45,46,47]. A survey was conducted on 50 volunteer Nigerian nursing mothers from three districts of Ogun State (South Western Nigeria) who provided samples of the foods they consumed most frequently along with samples of the breast milk they produced during the period of the study [46]. The analysis of the food samples revealed a high prevalence (93.75–100%) of AFB1, but the contamination levels (0.07–0.89 ng/g) were below the European Union (EU) Maximum Tolerable Limit (MTL) of 2 ng/g. Meanwhile, 82% of the milk samples they produced was contaminated with AFM1 at concentrations ranging between 3.49 and 35 ng/L, with 16% exceeding the EU MTL for infant milk formula of 0.025 ng/g [48], with a significant correlation (r = 0.33) between AFB1 intake and AFM1 excretion into milk. 

### 2.2. Feed as an Indirect Source of Human Breast Milk Contamination with AFM1

Feed contamination with AFB1 contributes indirectly, but significantly, to increase the levels of AFM1 in human breast milk [47,49]. A portion of the AFB1 ingested by lactating domestic animals is carried over as AFM1 into the milk they produce, which will in turn be transferred to breast milk of nursing mothers consuming such contaminated milk or its derivatives. The persistence of aflatoxins in processed dairy products owing to their resistance to technological processes, such as heat treatments and fermentation [50], represents an important contributing factor to the increase of AFM1 levels in the breast milk [40,47,51] fed to suckling infants, thereby exposing them to various health conditions.

Dairy product contamination with AFM1 vary greatly depending on the level of feed contamination with AFB1 and the rate of its transfer, i.e., the carry-over rate (COR), as AFM1 to milk. The COR has been intensively studied in domestic lactating animals where it was reported to vary depending on many factors, including the species, the breed, the stage of lactation, the season of the year, the feed composition, the health status of the lactating animal, the milk yield, and individual variability [22,38,39,40,41,52,53]. Different mathematical models have been proposed to predict AFM1 concentration in milk based on AFB1 intake by lactating animals via feed and, hence, conformity/non-conformity of the milk to the regulatory standards. This would in turn allow setting the appropriate MTL of AFB1 in feed to guarantee that the concentration of AFM1 in milk do not exceed safe levels. However, none of these models could apply to all cases and under all milk production conditions for the same lactating animal species. In most cases, the experimental data did not fit those calculated by the models developed for the same or similar experimental settings [52,54]. For example, according to a steady-state model developed by van Eijkeren et al. [55], the maximum COR of ingested AFB1 into AFM1 of 3.2% was about two-fold lower than that predicted by using other models [38,41,52]. Additionally, the application of this model to observed data from similar studies did not fit and have yielded AFM1 concentrations differing by a factor of 0.9 to 1.3 [54]. Regardless of modelling studies, experimental data reported mean COR values ranging between 0.032 and 6.2% depending on the animal species and the breed among other experimental parameters.

COR values for humans have not received enough research interest to provide a clear idea on the portion of dietary AFB1 converted to AFM1 and be transferred into breast milk. In one study on a limited number of lactating women (*n* = 5) and for a short follow-up duration (4 and 5 days), 0.1–0.4% of the ingested AFB1 was carried over to breast milk as AFM1 [56]. Another study demonstrated a significant correlation between the ingested AFB1 in 50 lactating mothers from Ogun State (Nigeria) and the excretion of AFM1 in breast milk [46] without calculating the carry-over ratio. Instead, it demonstrated that mothers fed on feed contaminated with AFB1 at levels varying between 0.16 and 0.33 ng/g have yielded breast milk containing 3.49 to 35 ng/L of AFM1.

Considering the experimental and model-calculated data on the carry-over of feed AFB1 into milk AFM1, there is a general agreement that the EU MTL of 5 ng/g for dairy feed can adequately prevent the levels of AFM1 in the milk of lactating animals from exceeding the EU MTL of 0.05 ng/g [55]. Nonetheless, reduction of the AFB1 MTL in dairy feeds to 1.4–4.0 ng/g were suggested to ensure higher guarantee for AFM1 levels in milk to be kept below 0.05 ng/g [38,41,53]. Conversely, the AFB1 EU MTL of 20 ng/g in feeds other than those destined for lactating animals resulted in higher frequencies of AFM1 levels exceeding the MTL of 0.05 ng/g in milk [41,51] but remained below the US Food and Drug Administration (USFDA) MTL of 0.5 ng/g [51,53]. It should be emphasized, however, that these MTL values remain too high for baby foods, which should be kept below 0.025 ng/g according to the UE regulations [48]. Moreover, the more relaxed the regulations for AFM1 MTL in milk, the higher is its intake by the nursing mothers consuming such milk and derivatives eventually leading to higher exposure of suckling infants.

## 3. Physiological Mechanisms of AFM1 Secretion into Breast Milk

During lactation, the AFM1 that accumulates in the liver from the dietary intake (dairy products) and from AFB1 carry-over is distributed via the bloodstream to the mammary glands where it is secreted with the milk constituents in the secretory epithelial cells lining the alveoli of the lactiferous lobules. AFM1 then follows the normal milk flow, which transits from the lumen of the secretory cell alveoli and their ductules to be excreted outside the mammary glands through the nipple via the collecting ducts (Figure 2a,b). However, the precise molecular mechanism(s) of AFM1 transport from the interstitial space (blood stream) across the mammary alveolar epithelium and its secretion by the epithelial cells into milk are poorly understood. Studies have reported that AFM1 crosses the epithelial cells from the basolateral side (blood stream) to the apical side (lumen) by passive transport [17,57,58,59,60,61]. The excretion of this toxicant by an active efflux transport involving a carrier protein, Breast Cancer Resistant Protein (BCRP), is being increasingly endorsed by scientific evidence (Figure 2c,d). Passive diffusion of AFM1 through the mammary gland and small intestine epithelia is also possible as suggested by its detection into blood serum and milk shortly (within minutes in the serum and few hours in milk) after its administration to lactating animals [17,57,58,59,60,61]. The lack of correlation between the excretion rate of AFM1 into milk and the secretion rate of major milk constituents (lactose, protein, and fat) using active transport by the alveolar secretory cells during lactation provides an additional argument for its passive diffusion through the mammary epithelium [57]. Although plausible and compatible with the lipophilic and small molecular weight nature of the molecule [62], such studies did not provide mechanistic evidence or specify the precise type of the passive diffusion used and whether or not it is bidirectional and concentration gradient-dependent. A more specific in vitro study showed that AFM1 translocates the epithelial cell-line Caco2/TC7 monolayer by passive paracellular diffusion [63] as evidenced by: (i) its very quick bidirectional passage (apparent permeability value of 105.10 cm/s × 10^−6^) from the basolateral side to the apical side and vice versa, (ii) its poor uptake by the epithelial cells; low concentrations of the aflatoxin were detected within the cells after 48 h of exposure to AFM1, irrespective of the dose (1000–10,000 ng/kg), and (iii) the ratio of absorption/uptake to efflux/excretion was lower than 2.0, an indicative value of a passive paracellular or transcellular diffusion, contrary to the active transport of xenobiotic whose absorption to efflux ratios are typically higher than 2.0 [64]. Although this study was carried out on epithelial cell-line of the small intestine, its outcome is likely to be valid for the mammary alveolar epithelium sharing many features with small intestine epithelium as physiological barriers for the absorption, bioavailability, and transport of nutrients and xenobiotics [8,15]. Nevertheless, the study raised questions calling for further investigations to confirm or refute this transport mechanism of AFM1 by using an appropriate cell-line model, such as the bovine mammary cell line BME-UV1 [30]. One of these questions was the asymmetric passage of AFM1 across the epithelial Caco2/TC7 monolayer with a greater flow from the basolateral side to the apical side direction than from the opposite one, which invokes a possible active transport.

As mentioned above, there is increased evidence that, like other xenobiotics and anti-cancer drugs, AFM1 is secreted into milk by an active transport using the drug resistance transporter protein (BCRP) or BCRP2 (Figure 2c,d). This carrier protein is the second member of the G sub-family of the ATP-binding cassette (ABC) efflux transporters, hence its second name “ABCG2”, the first member being the murine BCRP1, also designated ABCG2 murine, as opposed to ABCG2 human. To avoid confusion, we will be using henceforth BCRP2 as the human ABCG2, which was first isolated from multidrug resistant human breast cancer cells where it mediates resistance to the widely used drugs in breast cancer treatment anthracyclines [65]. BCRP2 was characterized to be an apical transmembrane protein expressed in humans by *ABCG* gene to protect blood-tissue barriers, such as intestines, mammary glands, and placenta against toxic drugs and carcinogens, including aflatoxins [8]. To fulfil this function, BCRP2 pumps its substrates outside the cytoplasm and prevents their accumulation eventually ensuring their elimination via bile/feces, urine, and milk. Structurally, BCRP2 is a-72 kDa protein comprised of a transmembrane domain (TMD) of 6 alfa-helices and a nucleotide binding domain (NBD) at the N-terminal where an ATP is hydrolyzed into ADP during transport. To be functional, two BCRP2 monomers associate in a homodimer forming a transmembrane channel that allows the efflux of a large spectrum of xenotoxic substrates, including different types of mycotoxins and their conjugates [8,66,67,68] (Figure 2d). During lactation, BCRP2 is upregulated with a concomitant increase in the yield of milk and excretion of mycotoxins [8,69,70,71,72,73,74]. This active efflux transport appears to be a common mechanism for all mycotoxins [62,70] with the notable exception of fumitremorgin C produced by *A. fumigatus* being a potent inhibitor of BCRP2 [75,76]. In particular, a direct link between BCRP2 and AFM1 efflux transport was confirmed in bovine mammary epithelial Cell (BMEC) [72,74] explaining the positive correlation between high milk yield and its content in AFM1 as was repeatedly reported [38,40,77,78]. This transporter protein, which acts as a natural protective means of lactating mothers against the toxicity of AFM1, among other aflatoxins, by reducing their systemic circulation, exposes the suckling infants to a higher health risks due to the weak detoxifying capacity of their immature liver [79,80]. As the multidrug and xenobiotic efflux transporters are beyond the scope of this review, for more information on the structural diversity, classification, and molecular functioning, the readers are referred to relevant literature reviews, e.g., refs. [67,68,81,82,83].

Unlike the documented active efflux transport of aflatoxins for their secretion into milk, very little is known about their uptake by the mammary epithelial cells. Figure 2c presents possible means for the uptake of AFM1 from the interstitial space by the mammary alveolar cells. Although poorly backed by experimental evidence, the passive transmembrane diffusion was often suggested to be a common uptake mechanism of aflatoxins down concentration gradient [36,78,84]. Once in the cytoplasm, the toxin can be excreted in the lumen by the same means, i.e., passive diffusion, and/or by an active transport system, e.g., BCRP2-mediated efflux transport. The use of active efflux transport jointly with the uptake by passive diffusion of AFM1 may explain its asymmetric flow across the epithelial monolayer demonstrated by Caloni et al. [63] as it is well recognized that efflux transport modulates the uptake of passive permeability drugs and toxicants [85]. Continuous evacuation of AFM1 from the cytoplasm through active efflux transport maintains the gradient concentration between the interstitial space and the cytoplasm driving the AFM1 flow in opposite direction of gradient concentration between the interstitial space and the lumen. Endocytosis (e.g., pinocytosis) and facilitated diffusion reported to allow xenobiotic uptake by the placenta epithelial cells [86,87,88], may also contribute to the uptake of AFM1 by the alveolar mammary epithelial cells. Moreover, endocytosis was reported to be a common mechanism for nutrient and xenobiotic uptake by the mammary epithelial cells [89,90], and aflatoxins, being xenobiotic toxicants, cannot be excluded from this transport mechanism. Meanwhile, it is being increasingly admitted that the uptake of xenobiotics is carried out by active transport using transporters that belong to the influx superfamily of Solute Carrier (SLC) proteins, the second superfamily of transporters beside the above-mentioned ABC superfamily of efflux transporters [85]. Among the SLC members, Organic Anion Transporting Polypeptides (OATP) 2B1 and Organic Cation Transporter (OCT) 1, located in the basolateral side of the mammary epithelial cells and whose expression is upregulated in the mammary glands during lactation [91] are likely to mediate the intake of AFM1. Indeed, OATP2B1 was associated with the uptake of ochratoxin A [92], suggesting that it may apply to all mycotoxins considering their structural relatedness and the low specificity of these carrier proteins. The implication of BCRP2 in the efflux transport of mycotoxins of various classes inferred that all mycotoxins are potential substrates [62,70]. Moreover, OCT1 was reported to mediate a cooperative vectorial transport of drugs with BCRP2 in the mammary glands where it allows their uptake from the interstitial space by the alveoli epithelial cells, whereas BCRP2 excretes them in the lumen [93]. OATP2B1 was also reported to carry out such a coupled transport with BCRP2 [94], suggesting that these influx transporters share the same substrates with BCRP2, including AFM1 shown to be a substrate of this efflux transporter [72,74]. Nevertheless, the precise mechanism(s) used by the mammary epithelial cells in AFM1 uptake and excretion into milk remain(s) to be explicitly clarified for better understanding of breast milk contamination with this aflatoxin.

## 4. Adverse Health Effects of AFM1 on Infants and Young Children

Infants can be exposed to AFM1 from their in utero life throughout the nursing period where breastfeeding can be the main or exclusive food source [95,96,97]. Under these conditions, the extent of exposure is highly dependent on AFM1 intake by the mother as well as the rate of its transfer to the baby through the umbilical cord, as a fetus, and then through breast milk, as a suckling newborn and infant. Such continuous exposure can cause teratogenicity, stillbirth, or miscarriage during pregnancy [97,98] or lead to physiological and neurological disorders that the child would suffer the consequences for the rest of his/her life, such as stunting, malnutrition diseases (e.g., kwashiorkor and marasmus), autism, nodding syndrome, and related cognitive disorders [99]. The highest incidence of these diseases is recorded in low-income and middle-income countries of the endemic regions, especially those of Asia and Africa under subtropical climate conditions [42,95,100].

Breast milk contamination with AFM1 is of serious concern to public health for three main reasons: (i) the potential high intake of AFM1 by neonates and infants who are fed mainly or exclusively on breast milk in case of a high contamination; (ii) the weaker detoxifying capacity due to their immature organs, mainly the liver, and higher metabolic activity; and (iii) the possible continued exposure after infancy and childhood to aflatoxins through various foods [101,102], which increases the risk for the onset of severe chronic endpoints, such as cancer, at younger ages. These conditions result in an overall increase in the susceptibility of infants and young children to aflatoxins by about three-fold more than adults [101,103]. Therefore, high exposure of pregnant and nursing women to AFM1 can be anticipated to increase the rates of disabilities, morbidities, and mortalities within a society resulting in a heavy economic and social burden.

### 4.1. Growth Impairment

Stunting, underweight, or wasting of children are the most documented growth impairments associated with the exposure to AFM1 during fetal life and infancy through umbilical cord blood and breast milk, respectively [98,104,105]. In the year 2020, 149.2 million (22%) and 45 million (6.7%) of the world children under 5 of age were affected by stunting (low height for age ratio) and wasting (low weight for height ratio), respectively [106]. In the sub-Saharan countries of Africa, notorious for the incidence of aflatoxins in their foods and feeds [107], the prevalence of stunting and underweight in children below 5 years of age were reported to be as high as 38% and 22%, respectively in 2015/2016 [108]. Although aflatoxin contamination is not the only etiology that would explain such a high incidence of growth impairment, its contribution cannot be overlooked.

From the public health standpoint, these growth disorders have life threatening or life lasting consequences on affected children. Stunting alone was estimated to cause the death of children by 14–17% and was considered to be an underlaying cause of poor cognitive, motor development, and educational performances; it was even suggested to be congenitally transmitted to the offspring [43,106]. Wasting, on the other hand, can be treated and weight gain can resume normally, but it increases the death risk depending on the severity or leads to stunting upon prolonged exposure or after recurrent episodes [95,106]. Despite the prevailing belief that under-nutrition, inadequate dietary intake, and gastrointestinal illness are the primary etiologies of growth impairment [109], it is now well established that exposure to mycotoxins, e.g., aflatoxins and fumonisins, in utero and throughout infancy and early childhood hinders the linear growth and weight gain [43,95,97,101,104,105,110,111,112]. This was further corroborated by the failure of interventions, such as the provision of appropriate education on nutrition and complementary feeding in addition to the implementation of proper water, sanitation, and hygiene (WASH) to improve significantly the linear growth and normal weight gain in undernourished children [113,114,115,116,117,118,119].

One of the earliest studies that demonstrated the causal relationship between growth impairment and exposure of children to aflatoxins was conducted in Benin and Togo of Western Africa [110]. In this study, the Z scores of the height-for-age (HAZ), weight-for-age (WAZ), and height-for-weight (HWZ) were determined in children below 5 years of age whose diet consisted of breast milk only (exclusive breast feeding), breast milk and weaning foods, breast milk and household foods, and weaning and household foods. When matched with aflatoxin-albumin (AF-Alb) adduct levels in the blood, the outcome revealed that the levels of AF-Alb were inversely related to HAZ and WAZ scores, i.e., the higher these levels were, the more severe the stunting and the underweight occurred in children. A subsequent longitudinal study on Gambian children showed that high exposure to aflatoxins from the perinatal period to the age of one year, as evidenced by high levels of AF-Alb adducts in the maternal, umbilical cord, and children blood, reduced significantly the height and weight gains in children from 6 months to one year of age [105]. Conversely, this study demonstrated that a reduction of AF-Alb levels in the maternal blood from 110 pg/mg to 10 pg/mg during pregnancy resulted in an increased height and weight gains of 0.8 kg and 2 cm, respectively by the children during the first year of growth. According to the authors, this effect stems from the maternal exposure to aflatoxins during pregnancy and extends throughout the first year of the children’s life, which has established the association between aflatoxin exposure during infancy and growth impairment. However, in the first 16 weeks after birth where the children were essentially breastfed, the serum AF-Alb was detected at low levels in only few infants (13 out of 115 babies: 11%), suggesting that were residual aflatoxin adducts from the maternal exposure [105]. Moreover, no direct causal effect between the growth impairment and the presence of AFM1 in the breast milk of the surveyed mothers was specifically demonstrated in this study, as the exposure to aflatoxins was measured indirectly by the blood levels of AF-Alb, which typically evokes AFB1 exposure [120]. However, since AFM1 can form albumin adducts, albeit at lower extents than does AFB1, AF-Alb can also indicate exposure to AFM1 [121]. In addition, AF-Alb levels in the children’s blood were shown to correlate highly with AFM1 levels in their urine [122], suggesting that the urinary AFM1 most likely originates from the diet (mother’s milk and/or complementary food) or from the maternal and cord blood in the case of neonates [98] rather than being an AFB1 metabolite due to the low AFB1-detoxifying activity in the young children [80].

The inverse relationship between AFM1 in breast milk and growth impairment in nursing children has been reported in different countries around the world. A study on lactating women from Tehran (Iran) showed that breast milk contamination at levels varying between 0.3 and 26.7 ng/kg (median of 8.2 ng/kg) was inversely correlated with height at birth but not with weight [123]. Another study conducted in the same country on exclusively breastfed children of 90–120 days old from the city of Tabriz showed that their exposure to an average AFM1-concentration of 6.96 ng/L (range of 5.1 to 8.1 ng/L) retarded both height and weight gain compared with children fed on AFM1-free breast milk [124]. The growth impairment of suckling children despite the generally low levels of AFM1 in breast milk was attributed to a chronic exposure starting in utero from the carry-over of AFB1 and/or AFM1 from the mother’s diet, as substantiated by the under-height at birth [123], and continuing exposure after birth through the maternal milk and/or weaning and household foods [95,123,125]. Table 1 summarizes the outcome of selected studies carried out on infants and young children in different countries from endemic regions to demonstrate the causal link between exposure to AFM1 in utero or via the mothers’ milk and growth impairment (stunting, underweight, or wasting).

The association of exposure to AFM1 from mothers’ milk with growth impairment in children from fetal to early life is well documented [110,123,124,126,127]. However, most of the studies were observational (cohort or cross-sectional) with less power as evidence and each of them has its own pitfalls and limitations (Table 1). To the best, they may make the causal link circumstantial rather than direct. In addition, no study has demonstrated the mechanism of action to elucidate how do aflatoxins, including AFM1, act to cause stunting, weight gain, or wasting in children. The demonstration of the mechanism of action is required by the IARC to consider the causal link being direct [131]. Therefore, there is a need for specifically designed studies to provide unequivocal clinical evidence for the association between AFM1 exposure and growth impairment. Longitudinal studies using cluster randomized controlled clinical trials on children intentionally given low doses of AFM1 appear to be best fit for such a purpose, but they are challenged with the dilemma of the ethical considerations for the use of human subjects. Prior to performing these trials, the experimental design should be described in detail and submitted for review and approval to national and international organizations concerned by the ethical questions of scientific research. Once approved, the research team should be committed to provide updates on a regular basis and to communicate to the authorizing body any amendments introduced into the protocol in due course. This procedure is intended to ensure that the study meets strictly the ethical principles, e.g., maintaining a favorable balance of risk/benefit and respecting participants among other provisions, while making a significant contribution to the advance of the scientific knowledge in the field. So far, only two studies have been done in this framework, and they have used AFB1 [119] and total aflatoxins [42] on partially breastfed children. None of them has specifically investigated the effect of AFM1 on breastfed children. The first of these studies was conducted in Kenya on infants who have been recruited before birth (starting from the fifth month of pregnancy) and followed until the age of 22 months for linear growth and the concentrations of serum AFB1-Alb [119]. Recruited mothers and their children were split into two cluster-randomized groups consisting of an intervention group, receiving aflatoxin-safe maize (<10 ng/g of AFB1) as complementary food, and a control group receiving a regular household maize known to be usually contaminated with higher levels of AFB1. The clinical trials of study were approved in 2013 by Institutional Review Board for Research of the International Food Policy Research Institute (https://www.socialscienceregistry.org/trials/105 (accessed on 10 June 2022)) and ended in 2016 revealing the absence of causal link between AFB1 intake and stunting at 22 months of age [119]. These results should be interpreted with caution, as the study suffered many limitations (Table 1), the most prominent of which was the high rate of follow-up loss and incomplete data collection in both the intervention and the control groups. Additionally, at the midline of the experiment (13 months of age), a-7% decrease in the stunting rate was observed but the exposure (serum AF-Alb) did not decrease. Conversely, at the endline (22 months of age), no improvement in linear growth was observed despite a significant decrease in serum AF-Alb. These results, which remain unexplained, suggest that interfering factors not considered in the design of the study, such as seasonal variation, environmental enteric dysfunction, immunomodulation, and hepatic metabolism of micronutrients, may have affected its outcome (Table 1). The second study of the kind (Cluster randomized controlled trial) being conducted on Tanzanian children was approved for clinical trials in 2019 by the Institutional Review Board (IRB) and the Tanzanian National Institute for Medical Research (NIMR) (https://www.clinicaltrials.gov/ct2/show/NCT03940547 (accessed on 12 June 2022)). In this study, children were recruited at birth and followed for linear growth and serum AF-Alb for one year, including 6 months of exclusive breastfeeding followed by mixed feeding (breastfeeding was not interrupted). Recruited children were split into two-group clusters: the intervention group were intentionally exposed to low doses of total aflatoxins (up to 5 ng/g) after the sixth month of age via complementary food on a continuous basis for 18 additional months. The research team has published yearly updates in scientific journals [42,43,44] and in the ClinicalTrials.gov website [132]; the final outcome is as yet to be disclosed. Although the latter study did not address AFM1 in breast milk, it may serve as a model for future studies on exclusively breastfed children in areas where nursing mothers are fed on staple foods highly contaminated with aflatoxins, hence likely to secrete AFM1-contaminated breast milk for the control cluster group. The intervention cluster randomized group should comprise mothers fed on aflatoxin-safe foods and their breast milk be tested for the absence or safe levels of AFM1.

### 4.2. Other AFM1-Related Health Issues

Several adverse health effects of aflatoxins on infants and young children have been proposed as standalone diseases or as possible underlying mechanisms of action for growth impairment. These include immunomodulation causing chronic immune activation; gastrointestinal diseases; nutrient maldigestion and malabsorption; and impaired bone growth and remodeling [133,134,135,136]. The common feature to all these diseases is that they are related to the disruption of the small intestine functions mediated by damaging its lining epithelium. Additionally, the pathological and clinical features associated with aflatoxin intoxications have been attributed to a sub-clinical condition known as environmental enteric dysfunction (EED) being an underlying cause of stunting and anemia [134,136,137]. Although EED is primarily associated with the ingestion of high load of fecal bacteria under poor water, sanitation, and hygiene conditions that characterize developing countries [138,139], it was reported to share overlapping pathways with aflatoxin-mediated diseases [135]. For example, like EED, aflatoxins were hypothesized to impair protein synthesis thereby promoting gastrointestinal infections and liver toxicity, ultimately leading to growth impairment [134,140].

Few and fragmentary studies have been done on the immunomodulatory effects of aflatoxins on children and their impact on liver toxicity and protein synthesis. Turner et al. [133] first reported that a high exposure of Gambian children to aflatoxins suppresses selectively their humoral immunity. A drastic reduction in salivary IgA titers of these children with a concomitant increase in the levels of serum AF-Lys biomarkers was observed. The authors concluded that a high exposure of children to aflatoxins compromises their immunity, which explains the frequency of their gastrointestinal infections and hence the burden of infant infection-related mortality in West African countries. However, the same study showed that the levels of serum AF-Lys did not correlate with the Cell Mediated Immunity (CMI) response of to test antigens (tetanus, diphtheria, *Streptococcus*, tuberculin, candida, *Trichophyton*, and *Proteus*), nor did it with the antibody response to rabies vaccines. Further studies are thus needed to substantiate the relationship between exposure of young children to aflatoxins and the immunomodulatory effects on one hand and to gastrointestinal infection-related mortality on the other hand.

A recent longitudinal study investigated the hypothesis that exposure to aflatoxins impairs protein synthesis in children with an emphasis on the proteins used as biomarkers of inflammatory reactions (C-reactive protein, α-1-glycoprotein) as well as other serum proteins directly or indirectly involved in growth development, such as transthyretin, lysine, tryptophan, and Insulin-like growth factor-1 (IGF-1) [135]. The study was conducted on 102 Ethiopian children (6–35 months of age), including 50 stunted children, living in an aeras with highly contaminated staple foods (>10 mg/g). The study aiming to relate chronic exposure to aflatoxins (AFB1, AFB2, AFG1, AFG2, and AFM1) to linear growth impairment showed no clear correlation between exposure to aflatoxins, separately or in combination, to the protein status, inflammation, or linear growth.

Exposure of children to aflatoxins has also been suggested as a possible etiology for the onset or the aggravation of the clinical manifestations of neurological disorders, such as autism and nodding. A cross-sectional study conducted in Italy on autistic children (*n* = 172) revealed a significant difference between the levels of AFM1 in their blood compared with control group of non-autistic children or those at a risk (genetic relatedness to autistic parents). Nevertheless, other studies suggested that, like EED, the neurological disorders associated with exposure to aflatoxins relate to intestinal lining damage and to microbiota–gut–brain axis imbalance (dysbiosis) [141]. This microbiota is known to play a central role in the regulation of the metabolism and homeostasis as well as in controlling the CNS functions via neural, endocrine, and immune pathways. Therefore, its disturbance causes inflammatory bowel diseases and systemic inflammation leading to the alteration of the central nervous system (CNS) functions as is the case in some neuropsychiatric conditions, including autism [142].

Malnutrition-related diseases, such as kwashiorkor, marasmic kwashiorkor and marasmus, faltering, stunting, nodding, organomegaly, and retarded mental and physical activities have been ascribed to chronic exposure of children below 5 years of age to aflatoxins since their in utero life [95,143,144,145,146,147]. However, the exact relationship between the exposure of children to aflatoxins and the occurrence of these diseases as well as the specific contribution of breast milk AFM1 to their onset has been poorly investigated and remain to be substantiated.

The occurrence of AFM1 in breast milk to different extents depending on the countries and the agroclimatic zones within the same country as well as the socio-economic conditions is well documented [46,148,149]. Due to the well-established toxicity of AFM1 in humans and animals [95], its occurrence in breast milk is of serious concern to the public health, as it affects individuals in their early life causing either immediate or delayed death or inducing lifetime disabilities. Therefore, there is an urgent need to address this issue at national level and globally regardless of the paucity of scientific evidence for a direct causal link between exposure to this toxicant and the claimed health effects it may cause to children in utero, during infancy, childhood, and even at later stages of their lives.

## 5. Risk Assessment of AFM1 on Infant and Children’s Health

### 5.1. Prevalence and Extent of Breast Milk Contamination with AFM1 around the World

Contamination of breast milk with AFM1 is well documented, although reports from industrialized countries are scarce or lacking. Table 2 summarizes the prevalence and contaminated levels recorded in various countries from different regions of the world in the last two decades. The table shows that AFM1 is consistently present in breast milk at varying levels, with most prevalence values ranging between 40% and 100%. Prevalence levels lower than 5% were occasionally recorded in some countries, such as Cameroon (4.8%), Iran (1.3 and 0.7%), Brazil (2.0%), and Guatemala (4.9%). In a recent study conducted in Angola, none of the 37 analyzed samples of breast milk was found to be contaminated [150], which may not be representative due to the low number of samples and hence remains to be confirmed in future studies.

Table 2 shows also that the levels of AFM1 in breast milk are highly variable among countries and even within the same country depending on many factors, such as the agroecological zone (AEZ), the season of the year, and the individual dietary habit of the nursing women and their level of education [46,47,149,151,152,153,154]. Irrespective of these factors, the highest AFM1 contents in breast milk were generally recorded in developing countries of Africa and Asia where they reached abnormally high levels. This was the case, for example, of Egypt [32,155], Sudan [156], India [157], Tanzania [128], and Cameroon [158] with the respective AFM1 contaminations of 5131.0 ng/L, 2561.0 ng/L, 1200.0 ng/L, 550.0 ng/L, and 625.0 ng/L. Such levels can be anticipated to mediate acute and possibly fatal toxicities in children belonging to the high-risk group of population. In view of such levels, it may be speculated that acute or fatal aflatoxin intoxications had happened in children of these countries without being diagnosed or reported. Alternatively, this may shed doubts on the accuracy of the analytical methods used to generate such data. Indeed, most of the methods used were semi-quantitative/screening, e.g., enzyme linked immunosorbent assay (ELISA) and thin layer chromatography (TLC), or have a low sensitivity and specificity, e.g., liquid chromatography coupled with fluorescent detector. In addition, the clean-up step was usually omitted in sample preparation prior to aflatoxin quantification. Nonetheless, high concentrations of AFM1 in breast milk samples from India were reported to be determined by one of the most advanced and reliable liquid chromatography techniques hyphened with mass spectroscopy [157]. Although likely to be accidental or not representative of the overall status of breast milk contamination with AFM1, such high levels question the safety measures taken to protect the health of the mother and child in the respective countries. They also clearly suggest that this issue is of serious concern to public health and whose risk should be adequately assessed and addressed worldwide. It can also be noted from Table 2 that AFM1 concentrations in breast milk have a general tendency to decrease within the last decade. Similar observation was made in a meta-analysis carried out by Fakhri et al. [149] who noted that a decreasing trend of AFM1 content in breast milk between the years 1985 and 2017. Such a trend can be attributed to the overall improvement of food hygiene and sanitation leading to reduced aflatoxin contamination of the mothers’ diet. The availability of increasingly accurate quantification methods for aflatoxins may have contributed to reduce the excess errors of aflatoxin estimations providing more realistic figures. Indeed, the recent advances in the analytical methods for mycotoxin quantification and the improved skills of operators has yielded more accurate and reliable data on aflatoxin contamination in breast milk.

**Table 2 ijerph-19-16792-t002:** Occurrence of AFM1 (ng/L) in human breast milk in different countries around the world from 2002 to 2021.

Country	Period of the Study (Month/Year)	Mean Concentration (Range)	Positive/Total Samples (%)	Analytical Technique	Clean-Up	LOD/LOQ (ng/L)	Reference
	Africa						
Egypt	04/2000–05/2002	300.0 (20–2090.0)	66/120 (55.0)	HPLC-FD	IAC	NS/NS	[155]
Egypt	05–09/2003	13.5 ^1^ (5.60–5131.0)	138/388 (36.0)	HPLC-FD	SPE	NS/NS	[32]
Egypt							[159]
Overall ^2^	NS	NS	248/443 (56.0)	HPLC-FD	SPE	4/NS	
January ^3^	NS	8.0 (4.2–108.0)	12/50 (24.0)	HPLC-FD	SPE	4/NS	
July ^3^	NS	60.0 (6.3–497.0)	24/26 (92.0)	HPLC-FD	SPE	4/NS	
Egypt	03–08/2010	74.0 (7.3–328.6)	87/125 (69.7)	ELISA	ND	NS/NS	[49]
Morocco	11–12/2017	5.8 (<LOD-13.3)	43/82 (52.4)	ELISA	ND	5/NS	[160]
Nigeria	04–06/2006	NS (<LOD-4000.0)	2/10 (20.0)	TLC	ND	2000	[161]
Nigeria							[46]
Ogun Central	06–10/2010	35.0 (42.7–92.1)	15/15 (100.0)	HPLC-FD	IAC	10/50	
Ogun East	06–10/2010	6.5 (<LOD-18.6)	12/18 (66.7)	HPLC-FD	IAC	10/50	
Ogun West	06–10/2010	3.5 (<LOD-5.4)	41/50 (82.0)	HPLC-FD	IAC	10/50	
Nigeria	NS	3.9 (2.0–11.0)	10/75 (13.3)	HPLC-MS/MS	SPE	2.0/NS	[12]
Sudan	NS	401.0 (13.0–2561.0)	51/94 (54.0)	HPLC-FD	LLE	13/NS	[156]
Kenya							[162]
Makueni	NS	8.46 (0.200–47.5) 10.83 (1.4–152.7)	85/98 (86.7) 4/18 (22.2)	ELISA HPLC	LLE NS	5/NS NS/NS	
Nandi	NS	0.02 (0.003–3.7) 0.06 (0.5–0.8)	38/67 (56.7) 2/21 (9.5)	ELISA HPLC	LLE NS	5/NS NS/NS	
Ethiopia							[151]
Overall	08/2017–03/2018	1.1 ^1^ (<LOD-143.3)	360/232 (64.4)	ELISA	ND	5/NS	
AEZ: Lowland	08/2017–03/2018	2.6 ^1^ (0.9–9.3)	101/120 (84.2)	ELISA	ND	5/NS	
AEZ: Midland	08/2017–03/2018	1.0 ^1^ (<5.0–1.9)	58/120 (48.3)	ELISA	ND	5/NS	
AEZ: Highland	08/2017–03/2018	0.6 ^1^ (<5.0–7.9)	73/120 (60.8)	ELISA	ND	5/NS	
Season: Wet	08/2017–03/2018	0.9 ^1^ (<5.0–2.9)	115/180 (63.9)	ELISA	ND	5/NS	
Season: Dry	08/2017–03/2018	1.2 ^1^ (<5.0–10.0)	117/180 (65.0)	ELISA	ND	5/NS	
Tanzania	11/2011–02/2012	80.0 (10.0–550.0)	143/143 (100.0)	HPLC-FD	IAC	5/NS	[128]
Cameroun	NS/1991-NS/1995	NS (5.0–625.0)	3/62 (4.8)	HPLC-FD	LLE	NS	[158]
Angola	8–9/2018–8/2019	ND	0/37 (0.0)	ELISA	ND	5.0/NS	[150]
	**Asia**						
UAE	01/1999–12/2000	560.0 ^1^ (123.5–940.0)	140/129 (92.1)	LC-UV	LLE	NS	[163]
Lebanon Fall-winter Spring-summer	11/2015–12/2016	4.1 (0.2–7.9) 5.0 (2.2–7.8)	41/43 (93.8) 63/68 (92.6)	ELISA	ND	NS/0.20	[47]
Turkey	NS/2006-NS/2007	5.7 (5.1–6.9)	8/61 (13.0)	HPLC-FD	ND	NS	[164]
Turkey	10/2007–03/2008	NS (61.0–300.0)	75/75 (100.0)	HPLC-FD	LLE	5/NS	[165]
Turkey	12/2008–04/2009	3.0 (1.3–6.0)	18/73 (25.0)	ELISA	ND	10/NS	[166]
Turkey	12/2014–06/2015	19.0 (9.6–80.0)	66/74 (89.2)	ELISA	ND	5/NS	[167]
Turkey	10–11/2017	6.4 (5.1–8.3)	53/100 (53.0)	ELISA	ND	5/NS	[168]
Turkey	06/2017–03/2018	3.1 ^1^ (<2.0–5.5)	NA/122	ELISA	ND	NS	[13]
Turkey	12/2018–06/2019	12.2 (5.0–23.2)	75/90 (83.3)	ELISA	ND	5/NS	[153]
Iran	11/2003–03/2004	9.5 (7.1–10.8)	8/132 (6.1)	ELISA	ND	5/NS	[169]
Iran	05/NS-09/2006	8.0 (<LOD-27.0)	157/160 (98.0)	ELISA	ND	5/NS	[123]
Iran	03–04/2007	7.00 (5.1–8.1)	20/182 (11.0)	ELISA	SPE	5\NS	[124]
Iran	NS	6.8 ^4^	1/80 (1.3)	ELISA	ND	5/NS	[170]
Iran	05–08/2011	20 ^4^	1/136 (0.7)	HPLC-FD	LLE	NS	[171]
Iran	06–07/2011	0.6 (0.1–5.0)	24/87 (28.0)	ELISA	ND	NS	[172]
Iran	04–10/2014	5.9 (2.0–10.0)	85/85 (100.0)	ELISA	ND	NS	[173]
Iran	09/2015–04/2016	14.7 (5.0–41.3)	98/150 (65.0)	ELISA	ND	10/NS	[174]
Iran	04/2016–01/2017	4.1 (3.2–8.8)	84/84 (100.0)	ELISA	ND	NS/25	[175]
Iran	06/2020–03/2021	7.1 (5.4–9.0)	39/100 (39.0)	ELISA	ND	5/NS	[176]
Jordan	02/2011–02/2012	67.8 (9.7–137.2)	80/80 (100.0)	ELISA	ND	NS/NS	[177]
Nepal	NS/2015–2017 ^5^	4.5 (LOD-316.0)	1355/1439 (94.0)	HPLC-FD	IAC	0.04/NS	[178]
India	07/2017–06/2018	13.7 ^1^ (3.9–1200.0)	41/100 (41.0)	UHPLC-MS/SRM	LLE	7.8/15.6	[157]
Bangladesh	10/2019–03/2020	4.4 (LOD-6.7)	32/62 (51.6)	ELISA	ND	4.0/NS	[179]
	**Latin America**						
Brazil	NS/2012	LOD > LOQ	2/100 (2.0)	HPLC-FD	IAC	0.3/0.8	[180]
Brazil	NS.	18.0 (<LOD-25.0)	5/94 (5.3)	HPLC-FD	IAC	4.0/21.0	[79]
Colombia	05–09/2013	5.2 (LOD-18.5)	45/50 (90.0)	HPLC-FD	IAC	1.0/2.0	[181]
Central Mexico Overall Winter Spring Summer	01–08/2014			ELISA	ND	0.92/2.79	[154]
		10.9 (3.0–34.2)	100/112 (89.3)				
		12.8 (3.8–20.9)	20/20 (100.0)				
		12.1 (3.0–34.2)	35/42 (83.3)				
		7.9 (3.2–18.9)	45/50 (90.0)				
Mexico	NS ^6^/2012	17.0 (5.0–66.2)	123/123 (100.0)	ELISA	ND	5/NS	[182]
Guatemala	06/NS–10/2014	13.0 (4.0–333.0)	14/286 (4.9)	HPLC/MS	IAC	NS	[183]
	**Europe**						
Italy	01/NS-12/2006	55.0 (<LOQ-140.0)	4/82 (4.88)	HPLC-FD	IAC	3/7	[152]
Portugal Overall Summer Fall Winter	NS/2015-NS/2016			ELISA			[184]
		7.4 (5.1–10.6)	22/37 (32.8)		ND	5/NS	
		8.0 (>LOD-10.6)	11/31 (35.5)		ND		
		6.8 (<LOD-8.9)	10/25 (40.0)		ND		
		6.7 (<LOD-6.7)	01/11 (9.1)		ND		
Serbia	01/NS-05/2013	10 (6.0–22.0)	10/10 (100.0)	ELISA	ND	1.5/5	[185]

^1^ Median value; ^2^ overall samples for one year (January through December); ^3^ Month of samples collection; ^4^ AFM1 was detected in only one sample; ^5^ Pre-winter season; ^6^ summer season. Abbreviations: LOD; limit of detection; LOQ: Limit of quantification; IAC: immunoaffinity column; SPE: solid phase extraction; LLE: Liquid-liquid extraction; NS: Not specified; ND: Not done; AEZ: Agroecological zone; ELISA: Enzyme-linked Immunosorbent assay; HPLC: High-performance liquid chromatography; FD—fluorescence detector; MS: mass spectroscopy; UHPLC: Ultra-high performance liquid chromatography; MS/SRM: Mass spectroscopy/selected reaction monitoring; TLC: thin layer chromatography; UAE: United Arab Emirate.

### 5.2. Exposure of Infants and Children to AFM1 via Breast Milk and Related Health Risks

Few studies have been carried out to determine the exposure of suckling children to AFM1 via breast milk. While none of such studies was done in North America to our knowledge, most of them were carried out in African, Latin American, and Asian countries where the incidence of aflatoxins in foods and feeds is notoriously high (Table 2). However, most of the available reports indicate that breastfed infants are less exposed to AFM1 than those fed exclusively or partially with complementary foods [110,123,126,127]. It has even been suggested that breastfeeding exerts a protective effect of children when their mothers’ diet is highly contaminated with AFB1 [12]. Nonetheless, it is well established that breastfed children from different countries of the world are exposed to various amounts of AFM1, with those of developing countries of Africa and Asia being exposed at significantly higher levels than those of industrialized countries. Table 3 shows that the lowest mean exposure values were recorded in Brazil, Morocco, and Portugal with mean values of 0.069, 0.35, 1.06 ng/kg bw per day, respectively, while the highest exposures were reported in Egypt, Nigeria, and UAE, with the respective mean values of 52.68, 73.00, and 80.00 ng/kg bw per day. I t appears that the latter exposures put children at high risk, as values ranging between 1.13 and 66.79 ng/kg bw per day (mean 11.06 ng/kg bw per day) correlated, although weakly, with stunting in exclusively breastfed Tanzanian infants [128].

Fakhri et al. [149] used a hazard index (HI) to estimate the health risks for children exposed to AFM1 from breast milk. The HI was defined as the ratio between the exposure and the malignant tumor dose (MTD_50_), according to Equation (1):HI = EDI/MTD_50_
(1)
where, HI is the hazard index; EDI is the estimated daily intake in ng/kg bw/day; and MTD_50_ is the dose of AFM1 that causes tumors in 50% of test animals. A MTD_50_ of 100 ng/kg bw/day was used for infants as the equivalent of a carcinogenic risk level of 1 in 100,000. An HI higher than 1 indicates that the exposed children are at a significant risk to develop cancer.

According to these calculations, the HI values ranged between 0.001 and 0.651 for children of 12 months of age and between 0.004 and 1.396 for children of a one month of age, indicating that only children of 1 month of age from UAE and Thailand of the 95th percentile were at risk [149]. By using the same approach, we found that only UAE children with the maximum exposure (worst scenario) were at risk, with an HI value of 3.780 (Table 3). No recent data is available for Thailand to compare the present situation with the former one.

It should be mentioned, however, that the above risk assessment methods were based on cancer development as the endpoint disease generally associated with chronic exposure to cumulative amounts of aflatoxins, which does not necessarily provide realistic insights on the health status of exposed infants and children while being still young. Alternatively, TDI values of 17 and 82 ng/kg bw/day were suggested to assess the risk of noncarcinogenic effects of aflatoxins that can affect suckling babies during infancy or childhood [186]. The TDI of 17 ng/kg bw/day was set by using a highly sensitive mouse strain as the experimental animal, while the TDI of 82 ng/kg bw/day was defined by using a less sensitive mouse strain. Considering the former TDI value, children from Egypt, Tanzania, Nigeria, Mexico, India, and UAE would be at risk, while only children from UAE would be at risk if the TDI of 82 ng/kg bw/day was considered (Table 3).

Although the above-mentioned risk assessments may be debatable, as they are based on single report or very few reports for each country, which does not necessarily reflect the real situation in the whole country, they strongly suggest that exposure of children to AFM1 from mothers’ milk cannot continue to be ignored from the public health authorities worldwide. Nonetheless, more elaborate and extensive studies are still required to provide accurate, realistic, and representative estimations of exposure to AFM1 in each country or region. Reports on the incidence of aflatoxins in breast milk continue to be published around the world, especially in developing countries, and they are expected to generate enough reliable data to perform meaningful risk assessment studies. Unfortunately, less attention has been given to this issue in industrialized countries where foods and feeds are well controlled to raise a real concern for the contamination of breast milk by aflatoxin carry-over. In addition, lactating mothers in these countries are generally educated and informed about the relevance of the diet and food hygiene for their health and that of their babies. Moreover, the availability of various foods at affordable prices allows them to scrutinize their diet according to their preferences and the medical and nutritionist advice during pregnancy and thereafter. Nevertheless, such a situation cannot be taken for granted due to the global trade of food commodities putting developed countries also at risk from the perils of aflatoxins despite their strict monitoring, implementation of laws, and use of recent scientific knowledge.

## 6. Regulations

At present, there are no specific regulations or standards for AFM1 in breast milk regarding its safe use as the main and highly recommended food for children, at least during the first 6 months post-partum. The lack of such regulations is mainly due to the absence of a threshold level of AFM1 above which mothers’ milk can harm the suckling children. The scarcity of risk assessment studies to provide reliable and science-based maximum limits that would help regulatory authorities setting statutory benchmark parameters for safety, such as the TDI and the Provisional Maximum Tolerable Daily Intake (PMTDI) is another weighting factor.

Moreover, enforcement of any regulation requires accompanying measures to inspect, test, monitor, and reprimand deviations/violations or condemn non-conform products. This is not technically feasible or reasonable in the case of the nursing women who feed their children with their own breast milk. However, the recent development of breast milk banks to supply medically or socially fragile babies (e.g., pre-term and abandoned infants) or whose mothers cannot breastfeed them, the situation may change in the future and food safety authorities worldwide will have to pay more attention to this relevant and yet neglected public health issue. The emergence of formal and non-formal breast milk sale is another factor causing growing concerns about the safety of breast milk supply in commercial or voluntary donation forms and its impact on the receiving children. The main attention is presently given to its microbiological quality, but testing for mycotoxins, including AFM1, should also be given due consideration.

Since commercial infant foods are the main substitutes for breast milk in the case of shortage or when it cannot be supplied by biological mothers to their children, MTL values set for infant formulae and follow-on formulae are often used as a benchmark to consider whether the breast milk is fit for the infants’ consumption. The MTL of 0.025 ng/g or mL set by the EU commission regulation [48] is the most frequently used for such a purpose. The more restrictive MTL of 0.01 ng/g or mL in force in Australia and Switzerland is also used when the highest degree of protection is sought [184]. Yet, none of these MTLs is binding for breast milk or has been demonstrated to guarantee its safety for suckling children. In addition, there is no explicit mention of breast milk among infant foods in the above-mentioned regulations.

It could be argued, however, that it is not necessary to regulate breast milk for contamination with AFM1 despite the many reports demonstrating the high incidence of this mycotoxin in it. Indeed, it is well established that breast milk remains the safest and healthiest food for babies and young children and that the nutritional and health benefits it provides outweigh largely the risk it poses because of its contamination with AFM1. Furthermore, several studies have demonstrated that the introduction of complementary foods in the children diet increases significantly their exposure to AFM1 compared with exclusively breastfed children [12,110,123,126,127]. To the contrary, a study suggested that breast milk plays a protective effect on children against AFM1 exposure [12]. Therefore, instead of implementing regulatory MTL for AFM1 in breast milk, it would be worthwhile to apply the As-Low-As Reasonably Achievable (ALARA) approach by taking preventive measures that would efficiently reduce the dietary exposure of mothers to this mycotoxin and its parent AFB1. This could be achieved, for example, by reducing or avoiding the consumption, at least during pregnancy and the nursing period, of foods known to be the main sources of these aflatoxins, such as maize, peanuts, and dairy products. Alternatively, lactating women can consume these food products after ascertaining their aflatoxin-safe contamination levels. In a study conducted in Kenya, an intervention group of pregnant women receiving aflatoxin-safe maize purchased from stockists had significantly less AF-Lys adducts in their serums than did the control group receiving home-grown and locally stored maize, indicating a lower exposure of the intervention group to aflatoxins [119].

## 7. Measures to Control AFM1 Levels in Breast Milk

The occurrence of AFM1 in breast milk is a complex and multifaceted issue that involves the degree of development of a country, the socioeconomic status of the mothers and their education level, the climatic zone where they live, the pre-harvest and post-harvest practices, the regulations of the country and the degree of their enforcement, the overall awareness among populations of the risk that aflatoxins represent to public health, etc. [148]. The incidence of AFM1 in breast milk and the health risks it poses to children are of more concern to developing countries than industrialized ones. Therefore, most of the suggested measures to restrain the exposure of infants and young children via the mothers’ milk target primarily developing countries, especially those of the tropics where the climate is favorable to aflatoxin-producing mold growth and toxigenesis.

Many actions have been suggested to bring the incidence and concentrations of AFM1 in breast milk to safe levels. These include the implementation of appropriate regulatory measures; the use of specific interventions to improve the diet of the mothers and their children; and the development of awareness about this insidious hazard on the mothers’ health and that of their babies.

Adequate and strictly implemented regulations of aflatoxins are increasingly advocated as an efficient means to control the occurrence of hazards in foods for quality insurance. In the case of breast milk, regulatory actions to reduce the incidence of AFM1 in breast milk can be considered at two levels:The level of the mothers’ diet to reduce the intake of AFB1 and AFM1 during pregnancy and nursing to the lowest possible levels. Unfortunately, the socioeconomic considerations in many of developing countries, regulations on aflatoxins are lacking, too permissive, or loosely implemented due to poor administrative and analytical capabilities. On the other hand, setting strict regulations on aflatoxins in foods and enforcing them rigorously to protect consumers in general and pregnant and lactating women, may not be a realistic solution under the present economic and technological conditions in most developing countries. This issue has long been debated; and its opponents raise the argument that it would lead to food shortage with more dramatic health effects on the mother and child, as most of the agri-food production would be condemned because of deviations from the standards [120].At the level of breast milk itself to distinguish safe from unsafe breast milk on the basis AFM1 content as is the case for infant foods. Although this can be feasible for breast milk banks where milks exceeding a regulatory concentration (e.g., 0.025 ng/g) can be discarded, it is not technically possible at individual level of lactating mothers. No official control of the mothers’ milk to feed their children can be realistically applied.

In view of the above limitations to set regulatory standards and control measures specific for the diet of pregnant and lactating women or the milk they produce, alternative measures have been suggested. The most recurrent suggestion was government actions aiming at increasing the awareness the mothers about the risk associated with the dietary aflatoxins, so they can avoid consuming foods potentially contaminated with high levels of aflatoxins, such as those known to be the most common vehicles for aflatoxins, e.g., maize, peanut, tree nuts, and dairy products, especially if they were stored for a long period (pre-harvest foods) or showing evident mould growth. The organization of training sessions for mothers before and after delivery to provide them with appropriate education on safe nutrition for them and their children is another suggestion to reach this goal. However, both actions prove to be of a limited efficacy on the improvement of the health of the children, especially their growth and development [113,114,115,116,117,118,119]. In fact, this issue cannot be managed separately from other factors influencing food contamination with aflatoxins and needs to be part of integrated aflatoxin management strategies. Ortega-Beltran and Bandyopadhyay [148] and Bandyopadhyay [185] recently reviewed a model of such a strategy centered on the biocontrol (use of atoxigenic *Aspergillus flavus* strains to compete with aflatoxin-producing strains) combined with other practices and actions. Such integrated management strategies are versatile and can be tailored to socioeconomic and geographic contexts of target countries. For example, the so-called aflasafe initiative first developed in the US by the US Agricultural Department-Agricultural Research Service (USDA-ARS) and then adapted to the Sub-Saharan African (SSA) countries combines the following main components [148]:Organization of campaigns to raise awareness and sensitization of women before and after delivery about the health effect of aflatoxins.Adoption of biocontrol practices in agriculture by using atoxigenic strains of *A. flavus*.Actions to improve harvest and post-harvest practices, including the storage structures and use of hermetic bags.Application of dietary interventions aiming to reduce the dietary exposure of mothers and children to aflatoxins.Monitoring crops for aflatoxin contamination and communicating the results to farmers.Modern market development.Technology transfer for manufacturing and distribution.Policy developments and capacity building to monitor, regulate, and control food contamination with aflatoxins.

Although such strategies were set by the United Nations (UN) in the 2030 Agenda to meet 17 specific goals for sustainable development for a global peace and prosperity, they will certainly reduce breast milk contamination with AFM1, since the mothers’ diet will be less contaminated with AFB1 and AFM1. Such interventions have reduced aflatoxin contamination of crops by >80% in SSA countries compared with crops produced by traditional agricultural practices.

Lastly, it should be emphasized that any effort to curb this insidious public health burden should involve as many as possible official, technical, economic, and social actors as individuals or as group of interest. In addition to the central role of the government and its bodies in developing and enforcing food safety regulations, all the other stakeholders, including non-government organizations, agri-food producers, scientists, and traders, should participate actively and in coordinated manner to ensure the safety of foods for the general population as well as for infants and children. However, achievement of this ultimate goal in developing countries on short run has been subject to controversies among food safety experts in regional and international organizations, see [187]. The most recurrent argument against quick and strict enforcement of stringent regulations is the risk for food shortages due to condemnation of a substantial part of foods produced locally for non-conformity and the increase of the production cost with consequent health risks of malnutrition and undernutrition and associated diseases. The availability of foods is as crucial to food security as food safety. Therefore, under the present socioeconomic conditions of developing countries, it would be more realistic to give priority to specific interventions targeting pregnant and nursing women to increase their food safety awareness and provide them as well as their children with appropriate health care and safe foods. Meanwhile, endeavor should continue to meet the other objectives on medium or long run.

## 8. Conclusions

Contamination of breast milk with AFM1 as result of carry-over from the diet is well established. The frequency of occurrence and the extent of contamination are highly variable from one country to another and within the same country, as they are dictated by many ecological, socioeconomical, and dietary factors. Lactating mothers from Low- and medium-income countries, especially the less educated and those who belong to the disadvantaged socioeconomic strata are prone to produce highly contaminated milk. Indeed, the highest frequencies and levels of breast milk contamination with AFM1 were reported in African countries, such as Kenia, Tanzania, Sudan, and Egypt.

There is a general agreement that breast milk contaminated with AFM1 put suckling children at a high risk to develop serious diseases that can be mutilating and leading to lifetime physical or mental disabilities. However, such risk has not been soundly quantified and the causal link between the exposure and the putative diseases has not been thoroughly demonstrated. Among the many health disorders claimed to be caused by AFM1 intake from breast milk as the exclusive diet or along with complementary foods, stunting has received the most attention. Nevertheless, studies carried out on this issue have yielded conflicting, uncertain, or controversial results. Further studies using appropriate experimental design and epidemiological approaches are still needed to provide convincing evidence for the link between exposure to AFM1 and stunting. Apart from stunting, studies to substantiate the array of putative diseases caused by AFM1 in children through breast milk, the extent of exposure that would induce each of them as well as the underlaying physiological mechanisms are scarce or lacking.

Irrespective of the insufficiency of scientific evidence to establish the causal link between AFM1 and the diseases it may cause in children, breast milk remains by far the best food for children to ensure a balanced and healthy development. Hence, its contamination by any microbiological or chemical hazard represents a serious threat to their health. On the other hand, the toxicity of AFM1 is beyond question and its presence in breast milk is de facto a serious concern, which should be properly addressed to prevent its presence in the milk or keep it as low as possible for the highest protection of breastfed babies. No specific action can by itself achieve such a goal, which is very complex and requires interconnected interventions involving multiple actors from different economic and social sectors. As a part of the world, every country should endeavor to achieve the sustainable development globally aimed to reduce poverty and promote health and well-being for everybody. Within this agenda, integrated management strategies to reduce food contamination with aflatoxins, including AFM1 in breast milk, have been suggested to provide aflatoxin-safe foods for the mother and child. The application of integrated management strategies that can be tailored to socioeconomic and geographic contexts of each country or region offers innovative and practical solutions for this multifaceted global issue that compromises food safety, and hence food security.

There are presently no specific regulations on breast milk contamination with aflatoxins that can be used to gauge the safety of breast milk for children nutrition. Even if they were issued, their applicability to nursing women would not be practically feasible. Therefore, strict implementation and monitoring of the existing regulations on aflatoxins in foods and feeds has been repeatedly suggested to be an appropriate alternative to reduce the incidence of AFM1 in breast milk. The Low levels of AFM1 in breast milk in industrialized countries compared with developing countries has mainly been attributed to the stringent regulations on aflatoxins and their strict implementation. The increased awareness of pregnant and nursing women about the impact of their diet on their safety and that of their children is another contributing factor to reduces breast milk contamination with AFM1.

Lastly, breast milk remains the unrivaled food for children and the prevention of its contamination with any microbial and chemical hazard should be given the highest priority to ensure the best conditions for the normal growth and development of children. The future of any country and the world as whole is intimately dependent on children as the pledge for success and prosperity and to meet the goals for sustainable development of the 2015 United Nations agenda. They should thus be carefully treated and cared about mentally and physically from the fetal life to the adulthood where they can be productive. The care and attention the children should receive can never be over-emphasized and their provision with safe and healthy nutrition since their early life plays a central role in such care.

## Figures and Tables

**Figure 1 ijerph-19-16792-f001:**
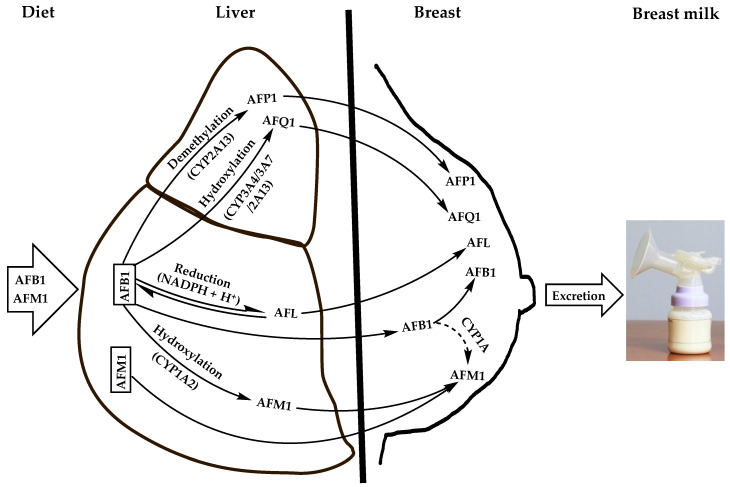
Natural contamination means of breast milk with dietary AFB1 and its major derivatives that are carried over to breast milk. Dashed arrow indicates an uncertain pathway that was demonstrated in vitro in bovine mammary cell-line (BME-UV1) to be likely carried out by a CYP1A isozyme.

**Figure 2 ijerph-19-16792-f002:**
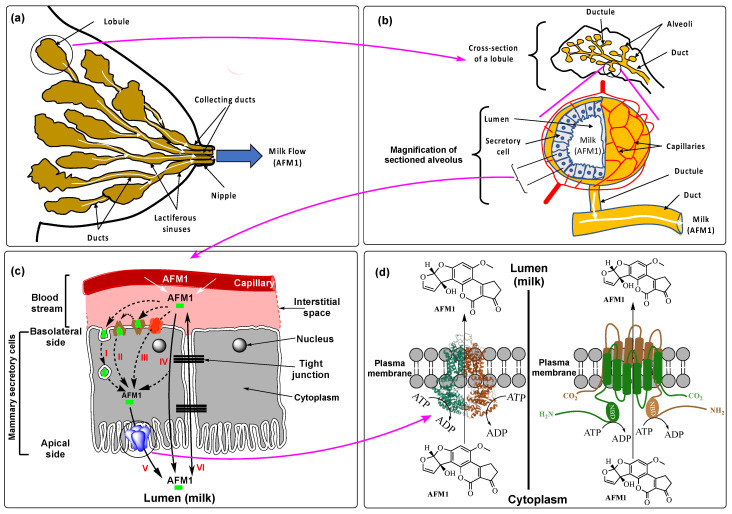
Fate of AFM1 after reaching the human mammary glands via the bloodstream to be secreted into breast milk. (**a**) Cross section of nursing mother breast showing secretory lobules where milk is formed and possibly contaminated with AFM1. The milk is then emptied into ducts to be conveyed into collecting ducts and ultimately excreted outside the mammary glands to feed suckling baby; (**b**) Cross-section of a lobule showing the constitutive alveoli as the basic units for milk synthesis and secretion and where contamination with AFM1 can occur (**top b**). Each alveolus is composed of secretary epithelial cells (**bottom b**) where milk constituents and the contaminating AFM1 are secreted into ductules and collected into a duct to be secreted outside the breast; (**c**) Two adjacent epithelial secretory cells of an alveolus depicting the possible uptake and secretion of AFM1 (green rectangle) into milk. The absorption can be done by endocytosis (I), facilitated diffusion (II), using specific transmembrane transporter protein of the influx SLC superfamily, e.g., OCT1 and OATP2B1 (red symbol) (III), or passive transmembrane diffusion (IV). After its uptake, AFM1 is excreted in the lumen with milk constituents through passive transmembrane diffusion (IV), specific BCRP2 efflux transmembrane transporter (V), and/or paracellular diffusion between two adjacent mammary epithelial cells (VI); (**d**) A schematic representation of the of efflux transport mediated by BCRP2 at the molecular levels showing BCRP2 assembled in a transmembrane homodimer, a cartoon drawing from PDB (6VXF) (https://www.rcsb.org/3d-view/6VXF/1 (accessed on 12 December 2021)) (**d-left**). The homodimer (monomers are in green and brown colors) is anchored in the plasma membrane with Nucleotide Binding Domain (NBD), the site of ATP hydrolysis, at the N-terminal in the cytoplasm, while the six alfa-helices of each monomer forming the transmembrane domain (TMD) are embedded into the phospholipid bilayer of the plasma membrane. The BCRP2 assembly forms a cylindric shape (**d-right**) with a central channel oriented so that it allows the passage of AFM1 from the cytoplasm to the lumen upon hydrolysis of two ATP molecules. White arrows in the mammary gland (**a**) indicate milk flow from synthesis in the lobule alveoli to excretion from nipples. Dashed arrows indicate hypothetical uptake means of AFM1 by the alveolar epithelial cells lacking empirical evidence for AFM1. Abbreviations: SLC: Solute Carriers; OCT: Organic Cation Transporters; OATP: Organic Anion Transporting Polypeptides; AFM1: Aflatoxin M1; ATP: Adenosine Triphosphate; ADP: Adenosine Diphosphate; PDB: Protein Data Bank.

**Table 1 ijerph-19-16792-t001:** Studies on the causal effect between growth impairment (stunting, underweight, or wasting) of children under 5 years of age and exposure to AFM1 from the mothers’ milk in selected endemic countries. The exposure to AFM1 from breast milk (ng/g or mL) is measured directly by estimated by the determination of AFM1 concentration in the milk (ng/g or mL) or indirectly by the determination of the children’s serum aflatoxin-albumin (AF-Alb) adducts (pg/mg equivalent of albumin) or AFM1 (ng/mL) in the serum or in the urine.

Country	Type of the Study	Period of the Study	Age in Months	Dietary Status	Mean (Range) of AFM1 Concentration in Breast Milk or Suitable Biomarker		Growth Impairment ^2^	Observations and limitations	Reference
					Biomarker ^1^	Breast Milk AFM1			
Benin and Togo	Cross-sectional	NS	9–60	Exclusively breastfed Partially breastfed Fully weaned (no breast milk)	32.8 ^3^ (5–1064)	-	Stunting (33%) ^3^ and underweight (29%) ^4^ Serum AF-Alb levels 30–40% higher than in control group	No direct causal effect: Confounding factors (high prevalence of malaria, diarrhea, respiratory infections) not ruled out No determinations of AFM1 in breast milk; use of AF-Alb biomarker is not specific for AFM1 Growth impairment rates increased with the contribution of weaning foods	[110,126]
Benin	Longitudinal (Prospective cohort)	February–October 2004	16–37	Partially breastfed Fully weaned	11.8–119.3 ^5^ (9.2–148.1)	-	Stunting High negative correlation (*p* < 0.0001) between AF-Alb and HAZ Weight gain not affected No significant: correlation between AF-Alb and WAZ	No direct causal effect: The height gain was more significantly affected in fully weaned children than in those partially breast-fed Use of AF-Alb is not specific for AFM1 Confounding factors not addressed	[127]
Iran (Teheran)	Cross-sectional	May–September 2006	<21	NS	-	8.2 ^3^ (0.3–26.7)	Stunting High correlation between AFM1 in the breast milk during gestation and the height of babies at birth.	No direct link between AFM1 in breast milk and stunting after birth. No significant correlation for the post-natal age Breast milk suggested to be safer than milk-based baby foods.	[123]
Iran (Tabriz)	Cross-sectional	March–April 2007	3–4	Exclusively breastfed	-	6.96 (5.1–8.1)	Stunting Significant (*p* < 0.01) adverse effect on HAZ compared to urban infants fed on AFM1-free breast milk Underweight WAZ not significantly affected by AFM1 content of breast milk (*p* > 0.05)	Confounding factors not eliminated: Deficiency of mothers’ milk in growth-promoting micronutrients Occurrence of gastrointestinal infections Lack of appropriate control.	[124]
Tanzania	Longitudinal (Prospective cohort)	November 2011–February 2012	<6	Exclusively breastfed Partially breastfed	-	0.07 ^2^ (0.01–0.55)	Stunting Significant (*p* < 0.05) inverse association between AFM1 intake and HAZ Underweight Significant (*p* < 0.05) inverse association between AFM1 intake and WAZ Wasting No significant association (>0.05) of AFM1 intake with WHZ	Good evidence for the link between AFM1 intake and stunting and underweight in infants (<5 months of age) fed on contaminated breast milk at levels exceeding the EU MTL of 0.025 ng/g.	[128]
Tanzania	Longitudinal (Prospective cohort)	NS	6–14	Partially breastfed	3.0–48.8 ^6^ (2.1–69.1)	-	No significant inverse association between AF-Alb and stunting, underweight, or wasting	No direct link Children were not exclusively breastfed AFM1 content was not determined in breast milk	[129]
Tanzania	Longitudinal (Prospective cohort)	November 2011–February 2012	<6	Exclusively breastfed Predominantly breastfed ^7^ Partially breastfed	-	0.08 (0.01–0.55) ^3^	Stunting and underweight 13% of exclusively breastfed infants (<3 months of age)	Lack of evidence for causal link Weak or insignificant association between growth impairment and AF intake in infants (<3–5 months) fed on different diets	[112]
Nepal	Longitudinal (Prospective cohort)	May 2010–February 2012	<36	Exclusively breastfed (<1 month) Partially breastfed (>1 month)	3.62 ^8^	-	No correlation between serum AF-Alb and LAZ, WAZ, or WLZ	Growth impairement was associated with confoundersrather than to AF exposure: Age and energy adjusted iron consumption correlated to LAZ Energy adjusted zinc consumption correlated to LAZ, WAZ, and WLZ	[130]
Kenya	Longitudinal (Cluster randomized controlled trial)	February 2013–November 2016	<24	Partially breastfed	18.1 (4.5–8.3) ^9^	-	Intervention with “aflatoxin-safe” complementary foods in chidren (0–22 months): No correlation between stunting and serum AF-Alb	Lack of sound evidence for causal link: Dietary interventions did not improve linear growth or the improvement did not correlate with aflatoxin exposure Interfering factors not considered in the study (see text). Unknown dose of AFB1 taken by children in the control group The contribution of mothers’ milk to the children diet and the extent of its AF-contamination not specified High rates of follow-up loss (20.6%) plus incomplete data collection (18.6%)	[119]
Turkey	Cross-sectional	June 2017–March 2018	<4	Exclusively breastfed	-	3.03 (2.59–3.82)	No significant correlation between WAZ and AFM1 in breast milk	The lack of inverse correlation between AFM1 intake and WAS in infants does not exclude the stunting effect at higher levels of exposure to AFM1	[13]

^1^ The biomarker is AFB1-Lys adduct the serum unless stated otherwise; ^2^ Growth impairment in terms of stunting (HAZ ≤ −2), underweight (WAZ ≤ −2), and wasting (HWZ ≤ −2) according to the WHO classification of malnutrition **(**http://www.who.int/nutrition/publications/childgrowthstandards_technical_report_1/en/); ^3^ Median value; ^4^ Percentage of infants and children below 3 years of age having developed this growth impairment; ^5^ Mean values per each of 4 studied villages; ^6^ Geometric mean values per each of 4 studied village and depending on the age of the children (at birth, 6 months, and 12 months); ^7^ infants receiving in addition to breast milk, plain water or water-based liquids; ^8^ geometric mean value; ^9^ Arithmetic mean and, in parenthesis, the geometric mean range in control children at about 22 months of age at the study endline (the period of August 2015 through October 2016);. Abbreviations: -: not determined; LAZ: Length-for-age z score; WAZ: weight-for-age z score; WHZ: weight-for-height z score; AF-Alb: aflatoxin albumin adduct measured in pg/mg equivalent of albumin; AFM1: aflatoxin M1; NS: not specified; ND: not detected; WHO: World Health Organization.

**Table 3 ijerph-19-16792-t003:** Exposure of children from different countries to AFM1 from breast milk.

Country	Age	Breast Milk Intake (mL/day)	Exposure (ng/kg bw/day)			Hazard Index (HI)			Reference
			Average	Min	Max	Average	Min	Max	
**Africa**									
Egypt	1–6 M	708	52.68	NA	NA	0.053	NA	NA	[49]
Tanzania	1 M	510	11.08	1.13	66.79	0.110	0.01	0.67	[128]
	3 M	690	11.94	0.81	58.96	0.119	0.008	0.590	
	6 M	770	10.91	1.08	34.90	0.109	0.011	0.349	
Nigeria	1–6 M	750–1300	73.00	NS	NS	0.730	NS	NS	[12]
Morocco	3–5 D	200	0.35 ^1^	NA	1.16	0.004	NA	0.01	[160]
**Latin America**									
Brazil ^2^	1 W	590	0.069	NS	NS	0.001	NS	NS	[79]
	1M	642	0.057	NS	NS	0.001	NS	NS	
	6 M	560	0.029	NS	NS	0.0003	NS	NS	
	12 M	452	0.019	NS	NS	0.0002	NS	NS	
Mexico	0–6 M	1980	5.08	1.52	20.18	0.051	0.015	0.202	[182]
	7–12 M	2350	4.68	1.31	9.98	0.047	0.013	0.100	
	7–12 M	2370	4.10	1.08	6.33	0.041	0.011	0.063	
	25–36 M	2020	1.81	1.25	2.28	0.018	0.013	0.023	
Central Mexico	0–6 M	750	2.35	0.92	6.28	0.024	0.009	0.063	[154]
**Europe**									
Portugal	NS	NS	1.06 ^3^	NS	NS	0.011	NS	NS	[184]
	NS	NS	0.86 ^4^	NS	1.25	0.009	NS	0.013	
**Asia**									
Lebanon	3–8 W	750	0.69	0.65	0.80	0.007	0.007	0.008	[47]
UAE	1 W	500	80.00	7.57	378	0.800	0.076	3.780	[163]
India	2–4 M	750	3.04	0.26	80.7	0.030	0.003	0.807	[157]

^1^ Median; ^2^ Exposure values are for female children; slightly lower values were recorded for male children; ^3^ Children of less than 7 kg body weight (bw); ^4^ Children of more than 7 kg/bw. Abbreviations: D: day; M: Month; W: week; UAE: United Arab Emirate.

## Data Availability

Not applicable.

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
