# Peer review of "Human Breast Milk Contamination with Aflatoxins, Impact on Children’s Health, and Possible Control Means: A Review"

_ijerph, 2022, doi:10.3390/ijerph192416792_

Round 1

Reviewer 1 Report

1.     The article is well written, especially the summaries of various publications in the field breast milk contamination with alfatoxins. What I see as a necessary for manuscript improvement is the reorganization of individual paragraphs, for example health effects are included in almost every chapter. So please reorganize the paragraphs and chapters so that they are more meaningfully organized. The text is well composed for review article.

2.     All figures and tables are not in a propriate order and also their names in the text are not appropriate, so please organise this in the right manner.

Figure 1 – In line 96 there is a mistake in ? made in the start of the line.

Line 172 - there is too much space in the parentheses.

Line 458- It is actually Table 1 – not the Table 5. This table is also well written but needs to be arranged more sensibly.

Line 544 – the start of the sentence is not meaningful.

Table 2 -  is not well organized, because there is the missing data in the list.

Line 686 – there is a yellow marked lock it. Please make it not marked.

Author Response

Reviewer 1

Dear reviewer
Thank you for the time and efforts to critically read the manuscript and for the suggestions to improve it. Please find below answers to your comments. The corresponding alterations are highlighted in yellow color in the revised manuscript

Kind regards
N. Benkerroum
Comment 1: The article is well written, especially the summaries of various publications in the field breast milk contamination with aflatoxins. What I see as a necessary for manuscript improvement is the reorganization of individual paragraphs, for example health effects are included in almost every chapter. So please reorganize the paragraphs and chapters so that they are more meaningfully organized. The text is well composed for review article.
Response 1: 
-    Numbering of sections and sub-sections were reorganised for more logical order and to increase the ease of the reader to follow the flow of the ideas. The main alterations were made in the heading 2 and its sub-headings 
-    A paragraph from the heading 7 about health issues caused by AFM1 in breast milk was moved to the sub-heading 4.1. In the present form, health issues are discussed in section 4 “Adverse Health Effects of AFM1 on Infants and Young Children” and section 5 “Risk assessment”. In the latter section health issues are discussed as specific endpoint.
Comment 2.     All figures and tables are not in a propriate order and also their names in the text are not appropriate, so please organise this in the right manner.
Response 2: Done 
Comment 3: Figure 1 – In line 96 there is a mistake in ? made in the start of the line.
In fact, the “?” was a caption to explain the question mark in the original figure beside CYP1A, the putative isoenzyme that would be responsible for the transformation of AFB1 into AFM1 in the mammary gland. As this is not soundly evidenced in humans, I added “?”. To avoid confusion, this was changed to a dashed arrow.
Comment 3: Line 172 - there is too much space in the parentheses.
Response 3: Extra space deleted
Comment 4: Line 458- It is actually Table 1 – not the Table 5. This table is also well written but needs to be arranged more sensibly.
Response 3: The table was modified to reduce its wordiness as much as possible to keep the clarity of the ideas in the hope it can be readable more easily
Comment 5: Line 544 – the start of the sentence is not meaningful.
Response 5: Sentence removed as it does not add much to the main idea of immunomodulatory effects of AFM1
Comment 6: Table 2 - is not well organized, because there is the missing data in the list. 
Response 6: A part of the table could not be seen in the “portrait” format. It was originally written in a “paysage” format but during downloading it turned automatically to portrait format. Hopefully this will nit happen when I will download the revised manuscript.
Comment 7: Line 686 – there is a yellow marked lock it. Please make it not marked.
Response 7: Corrected 

Reviewer 2 Report

This review article is a comprehensive account about contamination of human breast milk with aflatoxins derived from various sources and potential adverse health effects on infants and young children.

The article is well structured and covers in logically described chapters and sub-chapters the routes of breast milk contamination with AFM1, adverse health effects of AFM1 on children and risk assessment of AFM1 contamination of breast milk in view of exposure, in general and in various countries. Finally, current and potential future control measures are discussed. Conclusions are relevant and based on present research data highlighting the complex issue of aflatoxin contamination and control measures. Here , the authors could have noted more strongly the role and responsibility of all players in the feed-food chains, e.g. sensitization of producers and consumers, better quality control by processing industries and improved enforcement of regulations. 

The figures and tables are clearly presented and are a good bonus to the article.

There are a few editorial corrections to make as referred to below:

-Figure 1. Brest Milk should read Breast Milk

- lines 198 and 212: Cao2/TC7 should read Caco2/TC7

-line 566: AFB1, AFB1 should read AFB1, AFB2

-Table 2. line Angola 8-9/2028- should possibly read 8-9/2018

The authors could have added one recent study about contamination of breast milk with AFM1 and exposure of small children in Kenya. Reference is  Kang'ethe et al. Food Quality and Safety 2017, 1: 131-137.

Author Response

Reviewer 2

Dear reviewer,

Thank you for your time to critically read the manuscript and for your constructive comments to help improve the manuscript. Please find below responses to your comments. The corresponding amendment in the manuscript are highlighted in faint blue color.

Comment 1: This review article is a comprehensive account about contamination of human breast milk with aflatoxins derived from various sources and potential adverse health effects on infants and young children.

The article is well structured and covers in logically described chapters and sub-chapters the routes of breast milk contamination with AFM1, adverse health effects of AFM1 on children and risk assessment of AFM1 contamination of breast milk in view of exposure, in general and in various countries. Finally, current and potential future control measures are discussed. Conclusions are relevant and based on present research data highlighting the complex issue of aflatoxin contamination and control measures. Here, the authors could have noted more strongly the role and responsibility of all players in the feed-food chains, e.g. sensitization of producers and consumers, better quality control by processing industries and improved enforcement of regulations. 

Response 1: A paragraph was added to address this specific suggestion at the end of section 7 (lines 1016-1033)

The figures and tables are clearly presented and are a good bonus to the article.

There are a few editorial corrections to make as referred to below:

Comment 2: -Figure 1. Brest Milk should read Breast Milk

Response 2: Corrected

Comment 3: - lines 198 and 212: Cao2/TC7 should read Caco2/TC7

Response 3: Corrected

Comment 4: line 566: AFB1, AFB1 should read AFB1, AFB2

Response 4: Corrected

Comment 5: Table 2. line Angola 8-9/2028- should possibly read 8-9/2018

Response 5: Corrected

Comment 6: The authors could have added one recent study about contamination of breast milk with AFM1 and exposure of small children in Kenya. Reference is Kang'ethe et al. Food Quality and Safety 2017, 1: 131-137.

Response 6: Information added. thanks

Round 2

Reviewer 1 Report

The msnuscript is ready for submission.